# Insight into the Expected Impact of Sustainable Development in the Context of Industry 4.0: A Documentary Analysis Approach Based on Multiple Case Studies across the World

Wilian Jesús Pech-Rodríguez [ID], Eddie Nahúm Armendáriz-Mireles [ID], Gladis Guadalupe Suárez-Velázquez [ID], Carlos Adrián Calles-Arriaga and Enrique Rocha-Rangel *[ID]

Research and Postgraduate Department, Universidad Politécnica de Victoria, Ciudad Victoria 87138, Mexico; wpechr@upv.edu.mx (W.J.P.-R.); earmendarizm@upv.edu.mx (E.N.A.-M.); gsuarezv@upv.edu.mx (G.G.S.-V.); ccallesa@upv.edu.mx (C.A.C.-A.)
* Correspondence: erochar@upv.edu.mx; Tel.: +52-834-171-11-00

**Abstract:** Although industry 4.0 has gained increased attention in the industry, academic, and governmental fields, there is a lack of information about the relationship between this digital transformation and sustainable development. This work explores the concept of sustainability applied in industry 4.0 and the main advantages that this revolution incorporates into society. To this end, a conscientiously documented investigation was conducted by reviewing actual case studies or scenarios where sustainability was applied in different manufacturing industries, enterprises, or research fields worldwide. A critical and descriptive analysis of the information was performed to identify the main tools and procedures that can be implemented in the industry to address the triple bottom line perspective of industry 4.0, and the results are presented in this document. From the analysis, it was observed that currently, I4.0 has been mainly adopted to improve efficiency and cost reduction in manufacturing companies. However, since only a few enterprises embrace the social paradigm of I4.0, a significant gap in understanding and unbalance is visualized. Therefore, we conclude that there is a lack of information on social benefits and the barriers that must be overcome from the social perspective. On the other hand, this work highlights the importance of adopting industry 4.0 as a positive way to improve the performance of emerging technologies, such as fuel cells, solar cells, and wind turbines, while producing products or services with high efficiency and profitability incomes. For practitioners, this work can provide insightful information about the real implications of I4.0 from a sustainability perspective in our daily life and the possible strategies to improve sustainable development.

**Keywords:** industry 4.0; sustainable development; digital transformation

## 1. Introduction

At the beginning of the 21st century, society is experiencing the changes of the four-industry (I4.0) transformation, known as the digital revolution process. This revolution is helping to develop daily life smoothly due to the opportunities that this movement offers in areas, such as life quality, time and money savings, and environmental welfare [1,2]. This historical process is based on the profound use of cutting-edge technology, which improves the smart manufacturing processes, increasing production and capital [3,4]. On that basis, new technology companies have emerged to satisfy the market demand and promise to increase efficiency and improve operational times by incorporating digital networks which make production almost autonomous [5,6]. It is essential to highlight that I4.0 is not only focused on the improvement of margin saving in production, but is also committed to environmental and social well-being [7,8].

Thus, the term sustainability in I4.0 deals with social, environmental, and economic aspects (triple bottom line, TBL) to meet the needs of humankind, including the welfare and resources of future generations; see Figure 1 [9,10]. To this end, this transformation

is adopting technologies based on artificial intelligence, cyber-physical systems, big data, 3-D printing, and nano- and biotechnology [11–13]. Nonetheless, there is a broad debate on the true effect of industry 4.0 on sustainable development and the real impact of the triple bottom line philosophy on workplace and community [14]. From the literature review, it is fond that I4.0 is mainly adopted to increase revenues, and only a few enterprises are aware of the benefit of the TBL [15]. Hence, a comprehensive study needs to be conducted to clarify the benefits of I4.0 in the context of sustainability, especially those related to energy efficiency and waste management. Due to the global concerns about climate change, global warming, and energy efficiency, this paper first analyzes the environmental impact of I4.0. Although there is a lot of research focused on studying and analyzing I4.0 from a sustainability perspective, most of them are only aborded in a conceptual framework. Therefore, no tangible evidence of the benefits, challenges, and barriers are analyzed. Consequently, we focused our attention on actual case studies or real implemented scenarios where companies adopted I4.0 to analyze the environmental, economic, and social sustainability impact. The last part of this document discusses the interplay between these three dimensions, and the complex relationship is explained.

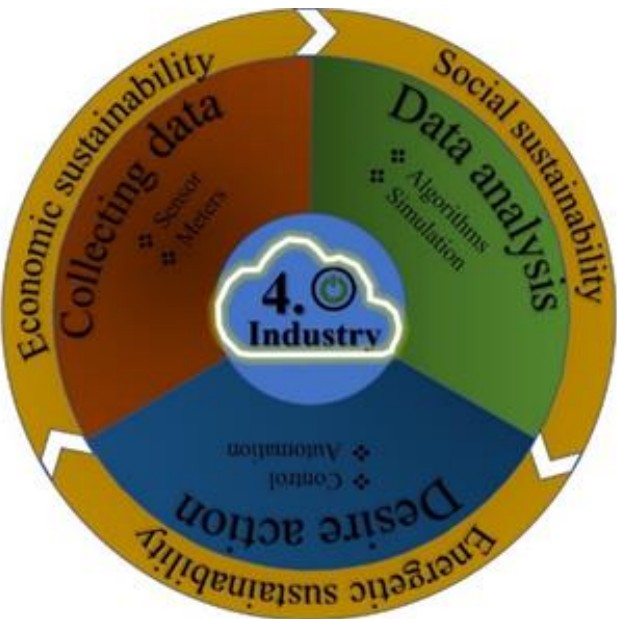

**Figure 1.** Sustainability aspects of Industry 4.0.

The main contribution of this work is the identification of I4.0 benefits through the in-depth analysis, considering the already adopted paradigm for some companies across the world. To do that, we provided a conscientious examination of each case and then discussed it from the triple bottom line of I4.0 perspective. We detected that the social dimension of I4.0 has an unbalance and this is because only a few enterprises have adopted this approach. Moreover, this work has highlighted the importance of I4.0 for the energy transition. Most eco-friendly developing technologies (fuel cells, solar cells, wind turbines, and batteries) are facing high costs of production and issues of traceability in the future. It is clear that the adoption of disruptive technologies can aim to achieve high power efficiency and enhance the life cycle of the aforementioned eco-friendly technologies.

Thus, the questions studied in this work are:

Research question 1: How is I4.0 affecting or supporting sustainable development?

Research question 2: What kind of tools can be adopted in I4.0 to achieve sustainable development?

Research question 3: What are the main barriers to sustainability in I.4.0, and how can they be overcome?

To respond to these RQs, a critical analysis was performed by reviewing the environmental sustainability of I4.0. The identification of the tools that the enterprises are using for the sustainable development pathway was achieved through the literature. Then, a method was developed to study the main barrier. The research objectives of the present document were:

Research objective 1: To determine the tools adopted for the I4.0 for sustainable development pathway.

Research objective 2: To identify the main barriers to the sustainable development of I4.0 and the strategies to overcome them.

The rest of the document is organized as follows: Section 2 presents the adopted methodology to collect and analyze data. Then, the analysis result of real implemented scenarios is presented in Section 3. Finally, a discussion about the impact of I4.0 is provided in Section 4, followed by the conclusions in Section 5.

## 2. Materials and Methods

### 2.1. Data Collection

This work was carried out by conducting a systematic literature review based on a case study of real scenarios where authors aborded the discussion and implications of I4.0 and sustainability by following the procedure proposed by Rosa et al. [12]. The systematic literature review was adopted to identify, analyze, and categorize all relevant articles or documents that engage the research questions.

A combination of keywords was used for the literature search. The first keywords were "case study" and "application". The other set of keywords includes sustainability, smart factories, artificial intelligence, manufacturing, sustainable supply chains, big-data, additive manufacturing, economic sustainability, "environmental sustainability", social sustainability, and "IoT" (including the term "industry 4.0"). The search was conducted in March 2022. The Boolean operator "AND" was used to combine the first set, and "OR" was used to combine the second. The search only considers documents written in English and published between 2018 and 2022. The initial search yielded more than 1300 documents on the Web of Science platform.

The second setup was the selection of relevant papers. It is worth mentioning that only case studies or real scenarios, where the implications of I4.0 were assessed, were considered. The documents were searched by selected criteria in the Web of Science database on the first attempt. Moreover, other sources, such as web pages, magazine business, and governmental reports, were consulted to consider all the advances in the area, and more than 800 sources were revised.

### 2.2. Data Analysis

Once the documents were collected, the authors performed a check by reading the abstract section of each paper, to eliminate the documents unrelated to I4.0. We want to highlight that only documents based on real case studies were considered. Surprisingly, it was observed that only 21 papers met the selection criteria. Furthermore, from the hand search, seven documents fulfilled the requirements. Although there is a lot of information, we only selected articles with enough data and evidence for the adoption and implications of I.40. The chosen articles from the Web of Science platform were crossed with the hand-selected search to identify duplicated samples. After that, the papers were thoroughly examined, and a text fragment that describes I4.0 and its implications was extracted; see Figure 2. Then, the information was divided into three blocks to assess the environmental, economic, and social framework impacts. The analysis of the report was carried out by three people with great knowledge in the area of sustainability and I4.0. This literature review aims to identify the case studies where I4.0 was implemented to improve waste, energy management, margin cost, and social impacts. Then, a correlation of data was performed to explain the effect of upcoming technologies on those areas.

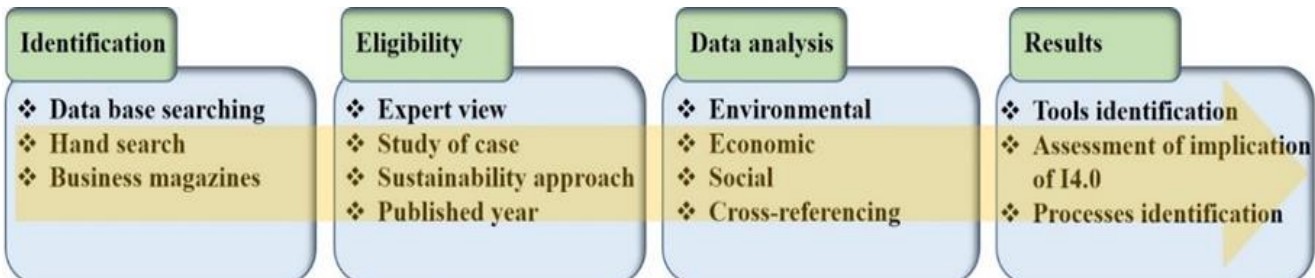

**Figure 2.** Step of the implemented Research methodology.

## 3. Results

The following sections explain the advantages and effects of I4.0 in each dimension. As mentioned before, this research was based on the extracted information from multiple cases studies implemented by several companies. Some cases were described in more than one dimension. In summary, a detailed analysis of the positive and negative consequences of I4.0 is presented.

### 3.1. Influence of I4.0 in Environmental Dimension: Environmental Sustainability

Nowadays, I4.0 is a trending topic in large and medium factories as a novel strategy to increase economic outcomes. In only a few cases, industrial digitalization is analyzed in the social and energy contexts [16]. Thus, in this first section, we consider the study of I4.0 from the environmental sustainability point of view. Guidance is performed to explore the broader implication of this transformation. It is worth mentioning that traditional manufacturing processes were developed to maximize profit. In fact, during their design, they do not contemplate the effect on the pollution rate, energy efficiency, and consumption of resources [17].

It must be pointed out that manufacturing is one of the largest energy consuming subsectors and is responsible for 33% of the consumed energy and 32% of the generated $CO_2$ [18]. Given this scenario, industrial transformation considers all complex aspects to guarantee the near safe future and help organizations and governments further understand the complex paradigm [19]. An essential facet of I4.0 sustainability is environmental management, and this can be studied from two different perspectives: the produced waste and the energy efficiency. Energy efficiency is a central topic of concern due to its increased cost and strict environmental policies [20]. The exponential growth of electrical demand and the depletion of fossil fuels have exposed the energy-related issues of future civilizations [21–24]. In the last years, the scientific community has made great efforts to develop new or improved technologies for electric energy production, such as fuel cells, solar cells, wind turbines, and so forth. I4.0 could be a feasible strategy for the energy transition, and the adoption of the Internet of Things (IoT), artificial intelligence (AI), big-data analysis, and digital twins could boost the improvement of eco-friendly technologies at a lower cost.

In the framework of suitable energy, the new industrial revolution can influence renewable energy and energy efficiency with a combination of management methodologies and technology, but this is a new approach and its impacts are still uncertain. Some organizations see digital strategies as a secondary effect on environmental sustainability, but people need to understand the role of energy efficiency and waste management in the industry. In contrast, this document highlights the impact of the adopted single technologies on environmental sustainability. The effectiveness and efficiency of equipment play a critical role in environmental sustainability. Thus, some concepts are defined to improve the understanding of energetic efficiency and the ability to identify the source of problems and their principal causal factors.

Overall equipment effectiveness (*OEE*) is a commonly used tool to assess manufacturing productivity [25] and is determined by the following equation:

$$OEE = (A \times Q \times P) \times 100 \tag{1}$$

where $A$ is the availability and represents the ratio between operating and planned production times, $Q$ is the quality, and $P$ is the performance efficiency (machinery ability in producing goods), and it is calculated by solving the equation:

$$P = \frac{\left(t_{cycle} \times Total\ count\right)}{OT} \tag{2}$$

where $t_{cycle}$ is the minimum time to achieve processing in one production unit. Giancarlo and coworkers derived a promising approach for energy consumption reduction in bath production processes by adopting a cyber-physical production system [26]. They use the concept of *OEE* to obtain quantitative data for energy losses. Figure 3 displays the time interval that is considered in the analysis of the *OEE*. An *OEE* of 100% means that only good items are being produced.

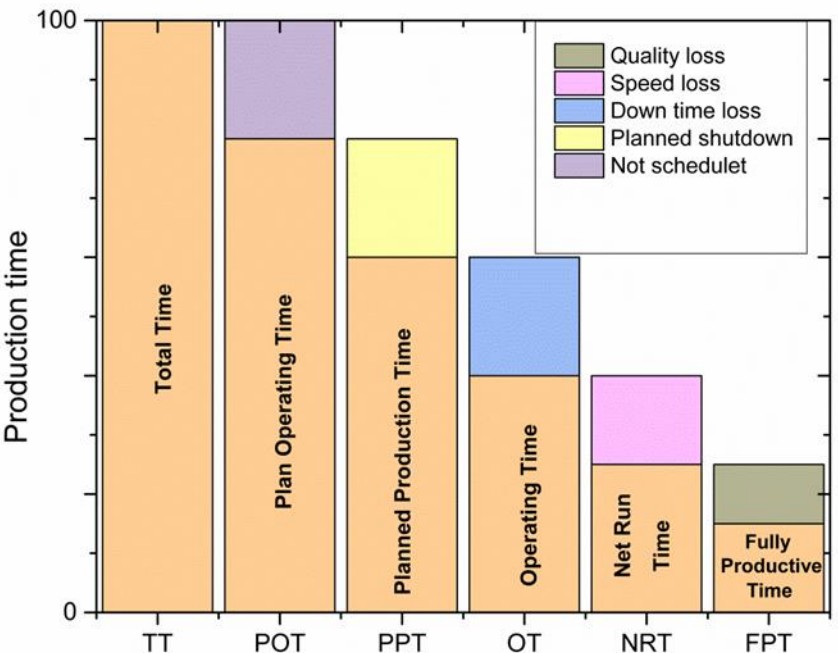

**Figure 3.** The overall equipment effectiveness time intervals.

They develop a case study of *OEE* for the automotive manufacturing industry to determine the causes and amount of energy waste. The process consisted in shaping a sheet metal strip by carrying out six operations with different molds. The manufacturing lines have robots to pick up the strips and transfer them from one press to the next stage. The energy consumption was determined by considering standby, average power to support, and average power in use, measured by two months. They found that plant failure represents the most significant issue because the energy consumption and $CO_2$ generation are increased. I4.0 can maximize the productivity and energy efficiency of manufacturing enterprises by getting real-time factory information, visualizing production data, and utilizing the equipment more efficiently.

Another example of the *OEE* improved under the adoption of I4.0, was performed in an Indian automotive parts exporter, Bharat Forge company [27]. The company uses a Nasdaq-listed IoT firm and PTC digital manufacturing solution to fully adopt digital transformation for two plants and have zero unplanned downtime in their processes. The company obtains a 15% *OEE* with excellent profit margins, and now they are implementing digitalization in all their machining shops to reduce cost and increase efficiency.

Another case study was reported in a cooperative ceramic company in Foshan, China [28]. The issue addressed by this company was the high energy consumption

and significant gas emissions. The production line use ball milling, moulding, and furnace sintering. They adopted a case-practice-theory-based (CPTB) model in a close loop structure to implement energy management. The energy consumption data collection was performed by I4.0 technologies, such as smart meters, RFID, RFID tags, and the cloud. After implementing the CPTB, the company experienced a reduction of 3% in energy consumption after one month.

Brozzi analyzes the environmental sustainability perception in a group of 65 manufacturing companies in Marche, Italy [29]. To do that, a triple bottom line (TBL) approach was adopted to assess the main factors in industry 4.0, and two research questions were set up in the sustainability direction [29]. From the data analysis, they observe that the current industry 4.0 does not see environmental sustainability as beneficial. The economic benefit prevails over social and environmental sustainability. Furthermore, the companies under study do not consider I4.0 as a tool for accomplishing environmental sustainability.

In separate research, Varela et al. evaluated the relationship between lean manufacturing and industry 4.0. in the Iberian Peninsula (Portugal and Spain) in the sustainability context [3]. The analysis of the collected data shows that lean manufacturing focuses on the actual company state. The latter is due to the non-holistic thinking that does not consider other sustainable paradigms. Meanwhile, the study reveals that environmental sustainability has the strongest factorial weight (1.482), followed by social sustainability. Finally, the authors state that the investigation is limited to the Iberian Peninsula and industrial companies, and further studies need to be performed in other countries.

On the other hand, Tan Ching and his research group used an interpretative structural model for energy sustainability, and they studied seven opportunities that I4.0 offers [30]. The mentioned work only studied the interrelation between the I4.0 and energy sustainability in terms of these seven functions. Still, there is no evidence of what the industry expects from this implementation. Digitalization offers the management of a flexible grid by changing its production in a smart way and, in this way, minimizing energy losses.

A Taiwanese electric scooter manufacturer performed exciting research on developing a new business model to face battery charging time issues in electric vehicles [31]. They used embedded technologies of the internet of things and cloud computing to monitor the battery performance in real-time. They gave the user information for battery replacement and predictive maintenance. The cloud data help companies determine their battery-charging stations and predict time and schedule. The latter is relevant research because one reason for the widespread practical use of batteries is the charging trouble. So, the implementation of technologies of the I4.0 can help in the structural transformation of sustainable energy sources.

The shipping industry has a substantial negative impact because it is a primary contributor to greenhouse emissions compared to other industry sectors. Emission released from raw material extraction in the shipbuilding phase were considered by using a life cycle impact assessment of a Panamax bulk carrier [32]. The study was performed considering a 76,300 deadweight Panamax carrier fabricated in Japan to evaluate the environmental impact of ship construction. The document explains that the extraction and production phases constitute the process that generates the account for $CO_2$, $CO$, and $NO_x$ gases. The global analysis was obtained by using a developed software (GaBi).

Monitoring electrical energy consumption through smart meters of the facilities is the first step towards efficient energy analysis. After that, the obtained data are integrated into the system and then analyzed to get the demand patterns, waste processes, and run simulations using adequate algorithms to obtain improved energy efficiency by making changes in different levels in the line process. For example, in Mexico, the high and medium voltage tariffs are divided into three blocks: base, medium, and peak. The cheaper tariff is the base and occurs from 0:00 to 6:00 h due to the small demand in the interconnected electrical grid. So, by the analysis of the electrical consumption of the factory and the results of simulations, the personal management can decide to move the most electrically demanding processes into the first tariff block. On the other hand, it has also been proposed

to include the use of removable energy in the process. Solar cells and wind turbines are the main choices because every location on Earth has at least sun irradiance or wind that can be used to generate electricity.

Jagtap and their colleagues undertook a real case study. They adopted the technology of the IoT to monitor in real-time the energy consumption of a beverage factory [33]. To this end, smart meters were used to collect the total energy demand of the production facility and the consumed energy of each food manufacturing process. The expert team found five areas of opportunity, and most of them were related to a re-engineering process, such as installing variable frequency drivers and replacing some old technologies with new ones. They obtain 807,081 kWh of annual energy savings. Another advantage of wireless networking technology is that the received data give the management personal knowledge to decide. Moreover, they can be assisted by expertise in data analysis to recognize areas of opportunity [34].

Simone et al. carried out a study to assess the potential of I4.0 in the decarbonization target promoted by the National Energy and Climate Plan (NECP) and the New Green Deal policies in Italy [35]. This research group analyzed a survey of about 300 companies implementing digitalization in their production process. The research questions were formulated to gather information related to the impact of the adoption of digitalization technology. The effects were assessed by considering the energy cost and the overall production efficiency of these companies. As a result, it was observed that the impact of I4.0 is higher in advanced automatization, such as the incorporation of variable speed motors and lighting. Incorporating automatization also reduces defective products, which avoids energy and raw material consumption. Hence, 50% of the companies have lowered costs related to personal involvement. The author concluded that many companies did not plan the I4.0 intervention as a sustainable energy plan. Nevertheless, the adoption of this new philosophy led to the reduction of energy, water, and waste consumption, and labour hours.

Similar to the other works, energy consumption is the second highest contributor to business expenditure compared to salaries and rentals. Adenuga proposes an energy efficiency analysis modelling system (EEAMS) to estimate the cost in a rail car manufacturing plant and provide a first-cut energy-efficient management strategy [36]. The latter can be achieved by continuously analyzing the big data generated in each process and applying smart software. The appropriate management offers the industry a direct way for certificates for energy management (iso50001). The proposed method is modelled considering the Minimum Efficiency Performance Standards (MEPS). The data collected from each sector can be modelled by neuronal networks, regression analysis, statistical analysis, or other tools to handle big data.

A model conceptualization diagram or causal loop diagram has been proposed by Hydayatno to fully understand the variables that support the implementation of I4.0 into the establishment of sustainable energy industrial development in Indonesia [37]. The studied input variables of the model were the industry growth rate, inflation rate, effect of technology, and emission factors. The output variables were industrial productivity, energy consumption, and gross domestic product. They expose the interrelationship between the three kinds of sustainability. This complex interaction was also observed by Ghobakhloo, who concluded that digital manufacturing technologies and digital twins offer a global sustainable effect [38].

With the aim to unleash the potential of solar cells, IoT has been used to remotely monitor in real-time the production, transmission, and distribution of the generated electricity. In this direction, Kusznier et al. conducted a study in a hybrid power plant at the Faculty of Electrical Engineering of the Bialystok [39]. The photovoltaic plant is composed of four sets of photovoltaic panels. One has a tracker to monitor the sun in a biaxial manner. The research group measured parameters, such as voltage, current, and temperature, and the data were stored in a local server to be analyzed. It is clear that I4.0 plays a critical role in

developing renewable energies because the data can be used to obtain the optimal solar cell parameters.

Considering the case studies mentioned above, it is evident that energy efficiency is the most studied topic under the environmental approach of I4.0. This is consistent with the result published by Beier and coworkers [40]. On the other hand, it can be observed that I4.0 has been used to deploy environmentally friendly energy sources. According to the literature, the solar cell and fuel cell are promising devices to generate sustainable energy. The adoption of I4.0 technologies can boost the efficiency of these devices and reduce their production cost. Waste management is a pending topic that needs to be explored in-depth under the perspective of I4.0 to assess the real impact. It must be mentioned that in some case studies, energy efficiency has been adopted to reduce the cost of energy consumption and not to address the environmental issues of global warming, air pollution, and water contamination. Table 1 summarizes each case study, and information on industry type, application, and country was added. It can be inferred that the manufacturing industry is the primary sector adopting the I4.0 from the environmental dimension. Meanwhile, environmental sustainability has been mainly applied in transport applications.

**Table 1.** Summary and information of case study: environmental dimension.

| Case Study | Industry | Application | Country |
|---|---|---|---|
| Energy Efficiency in Industry 4.0: The Case of Batch Production Processes [26]. | Manufacturing Industry | Automotive | Italy |
| Bharat Forge improves plant OEE by adopting Industry 4.0 solutions [27]. | Manufacturing Industry | Automotive | Germany, US and India. |
| A case-practice-theory-based method of implementing energy management in a manufacturing factory [28]. | Manufacturing Industry | Ceramic products | China |
| Industry 4.0 to Accelerate the Circular Economy: A Case Study of Electric Scooter Sharing [31]. | Auto industry (electrical vehicle) | Scooter | Taiwan |
| Cradle-to-gate life cycle assessment of ships: A case study of Panamax bulk carrier [32]. | Shipbuilding Industry | Seaborne transport | Japan |
| Real-time data collection to improve energy efficiency: A case study of food manufacturer [33]. | Manufacturing Industry | Food | India |
| Energy efficiency analysis modelling system for manufacturing in the context of industry 4.0 [36]. | Manufacturing Industry | Rail car manufacturing | South Africa |
| IoT Solutions for Maintenance and Evaluation of Photovoltaic Systems [39]. | Energy | Photovoltaic cells | Polonia |

### 3.2. Influence of I4.0 in Economic Dimension: Economic Sustainability

Strandhagen et al. studied the sustainability challenges in the shipbuilding supply chain and the impact of I4.0 [41]. The case company uses computer-aided design, but the design stage does not consider the performance of the ship operation phase, and the design phase is also facing issues of information flow. In this direction, the application of I4.0 technologies was suggested to improve competitiveness. A solution for these challenges was proposed considering the use of I4.0 technologies. Simulation, industrial robots, virtual reality, RFID, and IoT were applied to facilitate coordination and information transparency along the supply chains. From the report, it was observed that most of the proposed solutions were established to reduce the cost of operation process and time. For example, the IoT and radiofrequency identification were identified as a solution to increase the productivity of manufacturing logistic processes by tracking and tracing materials.

Supply chain management plays a critical role in modern enterprises because this involves creating and selling products, in other words, a pathway between the suppliers and the consumers. Supply chain management deals with the coordination of material flux, information, and cost and ensures product distribution at the right time. Allaoui and coworkers evaluated and validated a new platform for managing supply chains through a pilot program applied to food supply networks in Northwest Europe [42]. They use novel information and computer technology, namely the Collaboration Planning Tool (CPT), that aids in monitoring and planning supply chains across complex networks. They stated that it is necessary to incorporate robust real-time control to provide insight into the infrastructure and status of the supply chain. In this form, the system helps managers select the most suitable supplier.

Additive manufacturing (AM) is a technology that uses 3D printing to fabricate pieces bottom-up. AM enables the reduction of waste material and electrical energy consumption and has a relation between the circular economy and I4.0 [12]. Leendert studied additive manufacturing in two different scenarios: the aerospace sector and the construction sector [43]. The author found that I4.0 technology decreases energy demand by at least 5% and as much as 27% of the global demand, solidifying the outcomes for companies.

In this direction, it has been reported that the IoT, cyber-physical systems, and big-data applications in the industry have a tremendously positive effect on economic metrics. For example, a study was conducted in Brazil's plastic industry [44], where the research group measured the expected impact of I4.0 in this sector. The Fuzzy–Delphi method shows that additive manufacturing is not the most important I4.0 technology for plastic enterprises. The author claimed that the IoT and robots have a significant positive impact on profit because they improve production efficiency.

A complete study about asset traceability, from the perspective of I4.0, in the metal-mechanical sector was performed in a Mexican company that manufactures cargo transportation for the railway industry [45]. The study was focused on welding equipment because it represents the highest economic assets (6 million dollars) and represents 90% of the total processes. The Mexican company stated that 31% of their equipment is outside of the facilities and that this represents a severe problem for traceability, resulting in losses of about 1.4 million dollars. In this regard, the global positioning system (GPS) and radio frequency identification (RFID) was used as I4.0 technologies. As can be deduced, the company chose the GPS (this technology permits traceability at considerable distances). The company claimed that inventory reliability is close to 99%, and that this solution generates 553,634 USD in annual savings.

Another interesting case study was conducted in a German multinational instrumentation and control company located in Brazil [46]. This company fabricates equipment for measuring and monitoring sugar, alcohol, and chemical industrial processes, and they have an annual turnover of 150 million dollars. They have already implemented I4.0 technologies to improve information management, maintenance management, and production management. They implemented enterprise resource planning to control the process from the sales order to the shipping process, including the production management. Meanwhile, the maintenance management was improved by monitoring all manufacturing process variables such as energy, water consumption, and so on. They use an Arduino platform for remote control and create a supervisory system interface. The data analysis was carried out by an indicator called "index of the economy" that displays the percentage saving compared with the ageing process (without applying I4.0). They observed a substantial reduction in organizational costs.

The European community started a fascinating, innovative future-proof testing methods for reliable critical components in wind turbines project to assess the life cycle sustainability in large wind turbines [47]. This group uses the life cycle sustainability approach to improve capital investment and operational expenditures. To this end, emerging hybrid methodology and tools were adopted to model and assess the reliability of large and costly wind turbine components. The procedure consisted of physical and virtual testing to avoid

building expensive test benches. One study case was the development of a Gearboxes for 10 MW onshore turbine. They expect a 70% cost reduction with a 50% time-saving.

I4.0 can help the sustainable energy transition by providing valuable tools for the development and global analysis of emerging technologies. For instance, Gragiacomo et al. investigate the integration of hydrogen infrastructure for a new tramway route. Variables, such as speed, hydrogen consumption, station energy consumption, and the overall system efficiency, were studied in numerical modelling considering a hybrid power train (fuel cell interconnected with batteries). The authors proposed incorporating digital technological solutions to measure in real-time these variables and remotely control the main parameters to improve energy efficiency. Moreover, they proposed an interconnected system to monitor the hydrogen production process and guarantee the operation with a high level of safety [48]. Big data analysis could be used to schedule predictive maintenance and will create innumerable advantages.

Moreover, the wind market has been experimenting with a rapid growth in demand due to the benefits of clean electrical generation. I4.0 has also been implemented in the wind power industry to improve workshop automation, e.g., the Benckoff trademark developed the TwinCAT software to enhance the operational management of wind turbines [49]. This software interconnects the wind turbine station with central servers that have the ability to analyze in real-time the collected data, and simulations can be performed. This result well correlates with the theoretical works where IoT and digital twin technologies were implemented to improve wind farms and offshore wind turbines, respectively [24,50].

Stanley Black & Decker, Inc., a multinational power tool manufacturer in Reynosa, Mexico, turned to Cisco to find a real solution for their large-scale production and manufacturing complexity [51]. To identify the potential issues and how to deal with them, they integrate operational technology and information technology based on the networked factory floor. Production output, quality updates, and the effect of shift changes were the issues that the management wanted to improve. A real-time location system powered by Cisco and AeroScout was incorporated into the production line to gather information, and the benefits were high-quality end products delivered on time. After the digitalization was completed, the company immediately observed cost benefits and overall equipment effectiveness (an increase of 24%). Labour efficiency (reduction of waste time) was also improved by 10% compared with the production line operating without I4.0 technology. On the other hand, a second indirect advantage of digitalization was visualized; the company offers better service to customers by delivering the products on time.

It seems that Cisco enterprise is working hard to transform traditional manufacturing to the I4.0 perspective. Wunderlich-Malec Engineering industry adopted the HyperFlex (hyper-convergence infrastructure solution) platform that combines computation, storage, and networking [52]. The challenges for this company were to reduce the time and cost of automatization deployments and improve the performance and resiliency of the process. With hyper-convergence and multi-cloud environments, the company accelerates infrastructure deployments, reduces costs, and improves operational technology repeatability and reliability.

The benefit of I4.0 has been extended in different industrial sectors. For example, the fashion intelligence company Stylumia in Bangalore, India, was created to improve e-commerce, and was scaled up to USD 20 million in 12 months. They use machine learning technology and artificial intelligence to solve the problems of the fashion industry. [53]. This platform uses nowcast to extract customer expectations, products attributes, brands, and prices in real-time. The clients claimed a revenue increment from 200% to 400%.

A complete case study was adopted by Piramal Glass manufacturer focused on producing glass bottles for the pharmaceutical, cosmetic, and food industries [54]. The challenge of this company was the digitalization of operational process parameters because all the data were being manually collected and they did not have enough time to perform analysis. To this end, the manufacturer incorporates an army of IoT sensors and big-data analysis stored in a cloud provided by Azure to get real-time visibility. The data were collected from the

sensor of the high-speed conveyor line and then charged into the plant monitoring system on the cloud. The manufacturing operations have real-time visibility, and this permits uptime for the optimal performance of equipment, resulting in the reduction of energy costs. The benefit of I4.0 was an improvement of 2–3% in production efficiency, and the payback period was less than one year.

The Italian multinational company Cornaglia Group Spa, a supplier for the automotive industry, adopted the Smart Factory 4.0 and the M2M solution from Olivetti and Alleantia [55,56]. The Italian manufacturer adopts this approach to have control of production efficiency to understand the causes of slowdowns. After implementing the Smart Factory, the group can monitor the production in real-time because all their processes are digitalized. The main impact was the maximum revenue from the investment, reduction of production times, and predictive maintenance of the plant. Plaglieri was another company engaged in digital transformation thanks to Cisco and Alleantia by interconnecting its cosmetic packaging plant [57]. They use software that virtually connects the electronic devices (PLC and CNC) to control the production lines, namely the digital twin and cloud.

The three-year collaborative project CyberFactory#1 aims to solve the dilemma between manufacturing and security by using simulation, optimization, and resilience technologies. Cyber-range and digital twins were implemented to assess the cyber and physical trends of the interconnected digital factories. The project consists of delivering a realistic digital model of the factory, including real-time sensing, human–machine collaboration, etc. Then, resilience and enhancement were studied in electronic cards for the avionic production line. The author stated that by implementing statistical process control, a 20% of reduction in production defaults was observed [58].

Apparently, many manufacturing enterprises worldwide have been applying the philosophy of I4.0 to maximize their profits and obtain new business models to compete in the international market. It can be deduced that management directors are well informed about the benefits of the economic sustainability of I4.0, and they are investing to obtain a flexible digital company. Real-time monitoring and IoT were the principal technologies manufacturers incorporated in their process lines. On the other hand, developers have adopted the economic sustainability of I4.0 to improve the life cycle of wind turbines, solar cells, and fuel cells. It is important to highlight that cost is one main barrier to the widespread application of the aforementioned renewable devices. In Table 2, the case study considered in the economic dimension is presented. It can also be seen in this dimension that transportation is the sector that most adopts the benefits of I4.0. Furthermore, India, Italy, and the US are the major countries using the economic sustainability of I4.0.

*3.3. Influence of I4.0 in Social Dimension: Social Sustainability*

The purpose pursued by I4.0 is sustainability in the three dimensions, but a unbalance between these dimensions is observed from conceptual and case studies. For example, Saniuk and coworkers studied the social expectations and market changes in I4.0. In this sense, they undertook a survey consisting of three sections (question for customization, implementation of I4.0 in Poland, and benefits of I4.0). The analysis was performed using the CAWI (standardized computer-based internet interview). The results show threats related to technological unemployment because I4.0 requires people with technical abilities [59].

In this regard, in this section, we focus on a critical analysis of case studies focused on social sustainability to identify the main drawbacks of I4.0 in this perspective. Consumer co-creation plays a fundamental role in social sustainability because this e-commerce changes consumer perception and satisfaction. This strategy is based on exchanging ideas between customers, and enterprises help construct customer relations because they feel included in the paradigm of I4.0 [40]. Nevertheless, the successful adoption of this model requires the control of multiple factors to achieve affordable products with high quality and a high level of customization [60]. For mass personalization, companies need to have production flexibility that guarantees their profits, and the latter can only be obtained by industry 4.0.

Real-time monitoring conditions and networking architectures can reduce costs and errors by tracking and monitoring the entire process. A relevant project was performed by the Adidas store in Berlin where the clients could design their own sweaters through a real-time 3D scanning interface. This laborious task was addressed by adopting technologies, such as IoT, cloud, modular production, and real-time data update [61]. The benefits were customer satisfaction and a shorter supply chain. Regarding the difficulties of personalized manufacturing, factory-as-service (FaaS) has been implemented as an open manufacturing service [60,62]. This concept is based on interconnected micro smart factories (MSF) that use IoT and the digital twin. The FaaS is composed of the open manufacturing service platform, additive manufacturing, and modular design. A case study of actual MSF was achieved in a factory in Daejeon, South Korea. This connected SMF uses the IoT sensors and middleware to collect data, and finally, the base model is illustrated through the digital twin application.

**Table 2.** Summary and information of case study: economic dimension.

| Case Study | Industry | Application | Country |
|---|---|---|---|
| Sustainability challenges and how Industry 4.0 technologies can address them: a case study of a shipbuilding supply chain [41]. | Manufacturing Industry | Ship building | Norway |
| Decision support for collaboration planning in sustainable supply chains [42]. | Manufacturing Industry | Food | France, England |
| The effect of additive manufacturing on global energy demand: An assessment using a bottom-up approach [43]. | Manufacturing Industry | Aerospace and construction | France |
| Traceability in industry 4.0: a case study in the metal mechanical sector [45]. | Manufacturing Industry | Railway | México |
| The concept of the industry 4.0 in a German multinational instrumentation and control company: a case study of a subsidiary in Brazil [46]. | Manufacturing Industry | Control and monitoring | Brazil |
| Applying Life Cycle Sustainability Assessment to maximize the innovation potential of new technologies for critical components in wind turbines [47]. | Manufacturing Industry | Wind turbine | Finland |
| Insights for Industry 4.0 Applications into a Hydrogen Advanced Mobility [48]. | Energy | Train-Fuel Cell | Italy |
| The market challenge of wind turbine industry-renewable energy in PR China and Germany [49]. | Energy | Wind turbine | China, Germany |
| Leading tools manufacturer transforms operations with IoT [51]. | Manufacturing Industry | Tools manufacturing | México |
| Improving the repeatability and resiliency of industrial automation deployments with Cisco HyperFlex [52]. | Manufacturing Industry | Automation | US |
| Developing a framework of artificial intelligence for fashion forecasting and validating with a case study [53]. | Manufacturing Industry | Fashion | India |
| Case Study: How This Glass Manufacturing Company Leveraged Azure IoT To Get Real-Time Visibility [54]. | Manufacturing Industry | Glass | India, Us |
| Industrial IoT, Tim and Olivetti step up digital transformation of the Cornaglia Group [55,56]. | Manufacturing Industry | Automotive | Italy |
| Cornaglia Group Spa. Pisa, Italy: Alleantia [57]. | Manufacturing Industry | Automotive | Italy |

To obtain in-depth understanding of I4.0 from the perspective of social sustainability, a case study was conducted in an Italian manufacturing organization [63]. The company was a local sanitary producer that had around 200 employees. Because the work is labour-intensive, the management directors decide to adopt I4.0 technologies. To do so, automatic conveyors, robotic arms, and the IoT were integrated into the process line. Workers were considered in each step, and at the end of the reconversion, the employees were trained to acquire the skills and digital competencies. The first impact was in the social umbrella because the workers had safer and non-intensive tasks, resulting in fewer accidents and work illnesses. The second impact was from the economic perspective, and the company increased its production by 30%.

Another case study was achieved in the big manufacturing company Haier in Tsingdao, China, which produces items, such as refrigerators, washing machines, and air conditioners [64]. The company adopts an Internet-based platform to provide an intelligent home and industrial solutions. This company decomposes the original manufacturing enterprise into different platforms, called "PinTai". The idea of this decision was to create a new sustainable circle of enterprises, and they have a collaborative work with all the micro-enterprises and with others that do not belong to Haier. One advantage of this partition was the opportunity for new investors. The platform was a web portal, the front-end for the manufacturing communities. Under this topology, customers can order a product through the "Easy-Design" platform, and the design team inside Haier can compete with each other to get the design task.

A case study involving a soles manufacturing industry in the carousel packaging area (40% of workers) at the finish phase was performed in other work. Some of the identified tasks were: box preparation and classification, soles picking and counting, box lifting, and storing. The company has four packaging workstations managed by two operators. I4.0 was adopted by configuring a smart object framework composed of sensors and hardware to collect data. Posture and vital signs for each operator were analyzed, and they found that the first operator had worse results than the second. The analysis shows that operator one has more warnings for awkward postures [65]. The aforementioned case is promising because I4.0 can be applied to identify workers' physical behavior and perform a complete analysis to reduce labor risks.

A relevant study was conducted by the company Ionology (a provider of digital transformation and artificial intelligence) in a global manufacturing business [66]. The situation was that the material department of the global heavy machinery manufacturer was reluctant to move away from aluminium to composite materials. So, the first step was developing a peer-reviewed material science review. The company proposed AI to push the business into the research area and educate engineers who are uncomfortable with it. Ionology's solution was the creation of semi-supervised machine learning that generates data and compares materials. Thanks to the adopted technology, the company can perform visual simulation, and they can change the maintenance business model from preventive to predictive. The latter document is classified in social sustainability because Ionology has a comprehensive education model where all the staff were prepared to embrace the business potential of AI and it offers substantial competitive advantages in the social dimension. The latter is very important in the social paradigm because they train the current labor force in digital technologies. Table 3 displays the case study included in the social dimension. It can be concluded that the manufacturing industry is the principal sector that incorporates the benefits of I4.0 from its social perspective.

According to the previous revised case studies, social sustainability is the biggest challenge for the I4.0, and this has been previously reported from a conceptual point of view. Nara and coworkers published a critical study of the impact of I4.0 in the context of Brazil's plastic industry [44]. This group uses the triple bottom line approach of I4.0 for sustainable development. A fuzzy TOPSIS multi-criteria was adopted to clarify the effect of the leading technologies of I4.0 in the plastic industry of this country. They observed an imbalance of the TBL because the main advantages in the plastic industry

were related to economic metrics. A negative effect on social sustainability was detected due to the replacement of human beings with robots. In this sense, it is obvious that there is not clear evidence about the future work and job profiles of enterprises that adopt I4.0 transformation. In contrast, Naderi et al. make a reliable conceptual prediction of the operational management for I4.0 and its social return. They proposed using simulations and virtual reality to create changes in the production system. They stated that I4.0 has a social return, such as training human resources and generating documents related to I4.0 Euro region waste production, environment, and labor risk. We would like to mention that most studies that claim social improvements are studied in a conceptual framework.

**Table 3.** Summary and information of case study: social dimension.

| Case Study | Industry | Application | Country |
|---|---|---|---|
| Motives and barriers affecting consumers' co-creation in the physical store [61]. | Manufacturing Industry | Consumer co-creation | Sweden |
| Exploring the socio-technical interplay of Industry 4.0: a single case study of an Italian manufacturing organization [63]. | Manufacturing Industry | Sanitary ceramic | Italy |
| Industrial Cases Concerning Social Manufacturing [64]. | Manufacturing Industry | Mold manufacturing | China |
| IoT to Enable Social Sustainability in Manufacturing Systems [65]. | Manufacturing Industry | Shoes production | Italy |
| Artificial intelligence in business case study [66]. | Manufacturing Industry | Heavy machinery | Global |

Social sustainability is a scenario of great debate because some authors state that I4.0 brings advantages, such as new job opportunities and the possibility of labor force relocation, while others mentioned that job positions are vulnerable to automatization [46,67].

Lastly, it is very important to consider that an unforeseeable shock hit the global economy from the beginning of 2020, when the spread of the COVID-19 virus was first detected in China, then other countries around the world. In this context, Lepore et al. [68] suggest that I4.0 technologies can be a key tool for economic recovery by fostering the shift towards sustainable manufacturing. This requires measuring countries' readiness for I4.0 in order to guide policies in defining incentives to promote I4.0 and unleash its potential in the pandemic era. Their paper analyzes the preparedness and responsiveness of Italian regions regarding I4.0 concepts prior to the pandemic and identifies best practices supporting companies in adopting I4.0, focusing on those that incentivize sustainable practices. An assessment is made based on two dimensions: the willingness of companies to invest in I4.0 and favorable structural conditions. The assessment shows a group of alert regions versus a group of unprepared regions, indicating that "alert regions" are more likely to effectively manage and overcome the post-COVID-19 crisis through the adoption of digital technologies to enhance post-crisis resilience. It also indicates that any measure that fosters collaboration among stakeholders for the adoption of I4.0 will be essential.

Finally, we want to add that social media would be the cornerstone of social sustainability adoption because these platforms can disseminate the knowledge of positive benefits and they can be used as media for the interaction of producers and customers.

## 4. Discussion

The present document extracts the positive and negative aspects of I4.0 by overviewing the most important cases studies. It is noteworthy that this research does not intend to present a descriptive review, but rather an in-depth assessment of the adoption effects in I4.0, based on an analysis of actual cases. A first glance, I4.0 has an unbalance between their three sustainability dimensions. The latter is understandable because today's enterprises are concentrated on profit-maximization.

It is worth mentioning that, although energetic efficiency has been extensively studied and implemented as part of the environmental sustainability of I4.0, their integration is

mainly focused on cost decreases and not environmental concerns. One reason for this might be the few economic incentives and the lack of regulations or policies that could motivate the industrial sector to adopt environmental sustainability.

Table 4 summarizes the 27 actual case studies analyzed in the previous lines. The data were categorized according to the I4.0 approach, and information regarding technology and business novelty was also included. From Table 4, it can be observed that the enterprises adopted I4.0 to affect a specific dimension at the beginning. For example, the source 36 adopted I4.0 to have an environmental effect, but due to energy reduction consumption the company saw a cost reduction. Nevertheless, it is obvious that the three dimensions are interrelational and have convergence. On the other hand, survey findings (see Tables 1–3) demonstrate that Italy and India are the major countries pursuing digital transformation in all dimensions.

**Table 4.** Summary of case study and the analysis under I4.0 paradigm, technology and business novelty.

| Source | I4.0 Paradigm | I4.0 Technology | Business Strategy or Actions |
|---|---|---|---|
| [26] | Environmental | IoT, cyber-physical, sensors, robots | Energy and $CO_2$ reduction |
| [27] | Environmental | IoT, sensors | Energy and manufacturing cost reduction |
| [28] | Environmental | IoT, cloud, sensors, RFID | Energy reduction |
| [31] | Environmental | IoT, cloud, | $CO_2$ reduction |
| [32] | Environmental | IoT, | $CO_2$ reduction |
| [33] | Environmental | IoT, sensors, big-data | Energy reduction |
| [36] | Environmental and economic | IoT, networking, cloud, sensors, Big-data | Energy reduction |
| [39] | Environmental | IoT, cloud | Energy reduction |
| [41] | Economic | IoT, cyber-physical, sensors, simulation, robots, RFID, | Increase production |
| [42] | Economic | IoT, cloud, sensors, simulation, RFID, Big-data | Increase production |
| [43] | Economic | IoT, sensors, additive manufacturing, RFID | Increase production |
| [45] | Economic | IoT, cloud, sensors, RFID, GPS, | Increase production |
| [46] | Economic | IoT, sensors, | Profitability improvement |
| [47] | Economic | IoT, networking, cloud, sensor, simulation, digital twin, big-data | Manufacturing cost reduction |
| [48] | Economic | IoT, cloud, sensors, simulation, big-data | Manufacturing cost reduction |
| [49] | Economic | IoT, cloud, sensors, simulation, | Manufacturing cost reduction |
| [51] | Economic | IoT, networking, cloud, sensors, | Increase production |
| [52] | Economic | IoT, networking, cloud, sensors, big-data | Manufacturing cost reduction |
| [53] | Economic | IoT, networking, cloud, big-data | Profitability improvement |
| [54] | Economic | IoT, cloud, big-data | Manufacturing cost reduction |
| [55, 56] | Economic | IoT, cloud, sensors, big-data | Increase production |
| [57] | Economic | IoT, cloud, sensor, simulation, twin | Manufacturing cost reduction |
| [61] | social | IoT, networking, cloud, simulation, | Social welfare |
| [63] | social | IoT, sensors, robots | Risk and safety |
| [64] | social | IoT, networking, cloud, big-data | Social welfare |
| [65] | social | IoT, sensors, big-data | Risk and safety |
| [66] | social | IoT, AI | Human resource development |

It seems that economic sustainability has, in some cases, an overlap with energetic efficiency. This is obvious because energy consumption is one of the biggest expenditures for the industrial sector [36]. Thus, industrial managers implement energy sustainability to reduce production costs.

Figure 4 presents the percentage of cases in each dimension. The economic dimension is the most adopted philosophy, and this is easy to understand because the industry attempts to obtain higher incomes at lower labor shares.

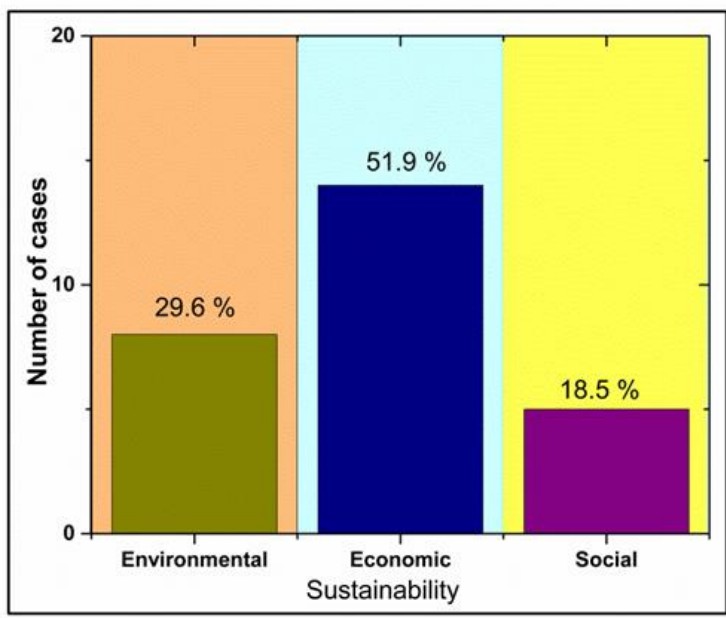

**Figure 4.** Distribution of the case studies under the I4.0 approach.

From the analysis, it can be seen that the social dimension has a significant gap in terms of understanding and estimating the effect of I4.0. Many theoretical pieces of literature describe I4.0 from a social perspective, but there is not enough evidence under which social circumstances will benefit. In fact, only five of the reviewed documents provide evidence of social welfare. Due to the lack of information, some consequences for social sustainability are not identified. One barrier to the global implementation of I4.0 is disinformation about the benefits of I4.0, and the other is an insufficient investment. As mentioned before, social media can unleash the power of I4.0 from a social perspective. Moreover, there is no doubt that I4.0 and its three dimensions represent a reliable platform for meeting humankind's sustainable development goals by adopting sustainable practices, green construction, and green sustainable supply chains in industry [69,70].

## 5. Conclusions

The present study endeavors to explain the advantages and implications of I4.0 from a sustainability perspective. To this end, a systematic literature review was undertaken by considering only case studies or real cases where manufacturing industries adopted the philosophy of I4.0. A set of keywords was used to search the information in the Web of Science database. Moreover, a hand search was also implemented to cover all types of documents. For practitioners, this document gives enough insight into the applications of I4.0 in daily life and the leading technologies that have been implemented to improve sustainable development.

✓ This work narrows the understanding and effect of I4.0 from a sustainability perspective in the form of a qualitative literature review. Because this document is based on the analysis of real situations, identifying the critical features can improve the understanding of the impact of the TBL of I4.0 fully. Furthermore, this document also contributes an analysis of what tools and processes have been applied to sustainability.

✓ From the social and economic paradigm case studies, it is clear that I4.0 has the potential to improve the efficiency and cost reduction of wind turbines, solar cells, and fuel cells, a very promising way to address the environmental and energy production issues.

✓ Additionally, we conclude that social sustainability in I4.0 has a large gap in terms of interpretation, and more systematic studies need to be implemented to close this unbalance. In order to overcome this gap, economic incentives and regulations for environmental and social dimensions need to be implemented to adopt sustainable development.

An important implication of our study is the unbalance in terms of social sustainability in I4.0. Thus, great efforts must be paid to create regulations and fiscal incentives that invite industrial managers to adopt social sustainability. Future work can address the social gap by conducting an exhaustive review of conceptual advantages and barriers of the social dimension. Then, a proposal list can be created in order to present ideas on how to potentiate the inclusion of the social dimension in I4.0.

**Author Contributions:** Conceptualization, W.J.P.-R. and G.G.S.-V.; methodology, W.J.P.-R.; validation, E.N.A.-M., G.G.S.-V. and C.A.C.-A.; formal analysis, W.J.P.-R. and E.R.-R.; investigation, W.J.P.-R., G.G.S.-V. and E.R.-R.; resources, W.J.P.-R., G.G.S.-V. and E.R.-R.; data curation, E.N.A.-M. and C.A.C.-A.; writing—original draft preparation, W.J.P.-R. and G.G.S.-V.; writing—review and editing, E.R.-R.; visualization, W.J.P.-R., G.G.S.-V. and E.R.-R.; supervision, W.J.P.-R. and E.R.-R. All authors have read and agreed to the published version of the manuscript.

**Funding:** This research received no external funding.

**Conflicts of Interest:** The authors declare no conflict of interest.

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
