# Peer review of "Insight into the Expected Impact of Sustainable Development in the Context of Industry 4.0: A Documentary Analysis Approach Based on Multiple Case Studies across the World"

_jmmp, doi:10.3390/jmmp6030055_

Round 1

Reviewer 1 Report

Dear authors, thank you for the opportunity to review your manuscript entitled: Insight into the expected impact of sustainable development in the context of Industry 4.0: a documentary analysis approach based on multiple case studies across the world. Further comments will explain the limitations of the work and suggestions for improvement:

The abstract is well described, my suggestion is to highlight the main findings from the literature study in the abstract.

The introduction is well structured, however, you need to better explain the need, aim, and context of your study. In the literature, we have a lot of papers that describe the relations between Industry 4.0 and Sustainable development, such as:

Industry 4.0 and sustainable development: A systematic mapping of triple bottom line, Circular Economy and Sustainable Business Models perspectives (https://doi.org/10.1016/j.jclepro.2021.126655)

Industry 4.0 and sustainability: Towards conceptualization and theory (https://doi.org/10.1016/j.jclepro.2021.127733)

Challenges and Benefits of Sustainable Industry 4.0 for
Operations and Supply Chain Management — A Framework
Headed toward the 2030 Agenda (doi.org/10.3390/su14020830)

You need to explain what is the novelty of your study. Please, at the end of the introduction add the structure of the paper.

The part of Materials and Methods needs to be improved. The methodology is a key issue in the paper and a reason for Major Revision. Even though the authors claimed that they used systematic literature review and bibliometric analysis methods, the description of the material and methods needs to be improved. I recommend the authors to read sources such as Guidelines for performing Systematic Literature Reviews in Software Engineering by Kitchenham: https://userpages.uni-koblenz.de/~laemmel/esecourse/slides/slr.pdf. There, one can find examples of the systematic literature review and learn how the paper should be organized, as well as how to increase the quality of the part of materials and methods.

The results part is well described. My suggestion to authors is to add some new papers on the relations between supply chains and sustainable development, such as:

1. Challenges and Benefits of Sustainable Industry 4.0 for
Operations and Supply Chain Management—A Framework
Headed toward the 2030 Agenda (doi.org/10.3390/su14020830)

2. Green Supply Chain Management and Environmental Performance:
The moderating role of Firm Size (doi.org/10.24867/IJIEM-2021-3-285)

3. A More Sustainable Supply Chain (https://hbr.org/2020/03/a-more-sustainable-supply-chain)

The discussion part is well organized and structured.

The conclusion part needs to be improved. In the first part of the conclusion please summarize your paper.  After that incorporate the main findings of the study. After that please describe the limitations and future implications. 

Author Response

REPLY TO REVIEWERS' COMMENTS

We are thankful to the reviewers for their important comments. By taking in account of their suggestions, some sections of the manuscript were modified for clarity (using the MS word track changes tool). Below, we give our rebuttals to the comment.

Reviewer 1

Dear authors, thank you for the opportunity to review your manuscript entitled: Insight into the expected impact of sustainable development in the context of Industry 4.0: a documentary analysis approach based on multiple case studies across the world. Further comments will explain the limitations of the work and suggestions for improvement:

Comment 1:

The abstract is well described; my suggestion is to highlight the main findings from the literature study in the abstract.

Response to comment 1:

We appreciate this comment by the reviewer. The abstract was modified by adding the suggestion.

Comment 2:

The introduction is well structured, however, you need to better explain the need, aim, and context of your study. In the literature, we have a lot of papers that describe the relations between Industry 4.0 and Sustainable development, such as:

Industry 4.0 and sustainable development: A systematic mapping of triple bottom line, Circular Economy and Sustainable Business Models perspectives (https://doi.org/10.1016/j.jclepro.2021.126655)

Industry 4.0 and sustainability: Towards conceptualization and theory (https://doi.org/10.1016/j.jclepro.2021.127733)

Challenges and Benefits of Sustainable Industry 4.0 for

Operations and Supply Chain Management — A Framework

Headed toward the 2030 Agenda (doi.org/10.3390/su14020830)

You need to explain what is the novelty of your study. Please, at the end of the introduction add the structure of the paper.

Response to comment 2:

Thank you for this comment. The introduction section was modified by adding the contribution of this work. It is worth mentioning that the result part of our work only considers a case study, in other words, real cases where industry adopts the I4.0 and measures the advantages or implications.

Comment 3:

The part of Materials and Methods needs to be improved. The methodology is a key issue in the paper and a reason for Major Revision. Even though the authors claimed that they used systematic literature review and bibliometric analysis methods, the description of the material and methods needs to be improved. I recommend the authors to read sources such as Guidelines for performing Systematic Literature Reviews in Software Engineering by Kitchenham: https://userpages.uni-koblenz.de/~laemmel/esecourse/slides/slr.pdf. There, one can find examples of the systematic literature review and learn how the paper should be organized, as well as how to increase the quality of the part of materials and methods.

Response to comment 3:

We thank you for this comment. We agree that there are some issues with the methodology description. Therefore, the materials and methods section was modified, and information was added to improve clarity.

Comment 4:

The results part is well described. My suggestion to authors is to add some new papers on the relations between supply chains and sustainable development, such as:

  1. Challenges and Benefits of Sustainable Industry 4.0 for

Operations and Supply Chain Management—A Framework

Headed toward the 2030 Agenda (doi.org/10.3390/su14020830)

  1. Green Supply Chain Management and Environmental Performance:

The moderating role of Firm Size (doi.org/10.24867/IJIEM-2021-3-285)

  1. A More Sustainable Supply Chain (https://hbr.org/2020/03/a-more-sustainable-supply-chain)

Response to comment 4:

We thank you for this comment. The recommended documents were added to the manuscript

Comment 5:

The discussion part is well organized and structured.

The conclusion part needs to be improved. In the first part of the conclusion please summarize your paper. After that incorporate the main findings of the study. After that please describe the limitations and future implications.

Response to comment 5:

We appreciate your comments. The conclusion section was re-writer considering your suggestions.

Reviewer 2 Report

** Line 45, statement is missing something; correct
** Line 51, change "in" to "on" for better Englissh
** Line 66: statement is missing something; "is affecting is supporting" is not good. maybe need "and" correct
** section 2.2: systematic review details are missing. you need to report the number of citation at identification phase (Figure 2) for each method (database, hand, and business); how did you merge the results and indetfy duplicates; repeat the same for Eligibility phase; the resulting number after Eligibility screening
** Number equations in lines 154 and 159
** which industries did you use for your case studies. which applications did you use for our case studies. which countries did you use for our case studies. Section 3, authors reporting case studies in auto mfg industry in India, ceramic company in China, others in Italy, Portugal, Spain, Taiwan, etc. add a table, similar to Table 1, summarizing these three questions to make it easy for readers.
** generally systematic review papers include 100s of references to be accurate

Author Response

REPLY TO REVIEWERS' COMMENTS

We are thankful to the reviewers for their important comments. By taking in account of their suggestions, some sections of the manuscript were modified for clarity (using the MS word track changes tool). Below, we give our rebuttals to the comment.

Reviewer 2

Comments and Suggestions for Authors

Comment 1:

** Line 45, statement is missing something; correct

Response to comment 1:

Thanks for your comments. The manuscript was modified according to your suggestion.

Comment 2:

** Line 51, change "in" to "on" for better Englissh

Response to comment 2:

Thanks for your comments. The text was modified according to your suggestion and by taking into account other reviewers' comments.

Comment 3:

** Line 66: statement is missing something; "is affecting is supporting" is not good. maybe need "and" correct

Response to comment 3:

Thanks for your comments. The line was corrected.

Comment 4:

** section 2.2: systematic review details are missing. you need to report the number of citation at identification phase (Figure 2) for each method (database, hand, and business); how did you merge the results and indetfy duplicates; repeat the same for Eligibility phase; the resulting number after Eligibility screening

Response to comment 4:

Thanks for your comments. We agree that the methodology section has some missing information. So, the section was modified according to your suggestion and by taking into account other reviewers' comments.

Comment 5:

** Number equations in lines 154 and 159

Response to comment 5:

Equations were numbered.

Comment 6:

** which industries did you use for your case studies. which applications did you use for our case studies. which countries did you use for our case studies. Section 3, authors reporting case studies in auto mfg industry in India, ceramic company in China, others in Italy, Portugal, Spain, Taiwan, etc. add a table, similar to Table 1, summarizing these three questions to make it easy for readers.

Response to comment 6:

Thanks for your comments. In order to enhance clarity, we incorporate three tables that include information about industry type, application, and country.

Comment 7:

** generally systematic review papers include 100s of references to be accurate

Response to comment:

The reviewed papers were more than 2000, and the methodology section added the information. Because most literature is based on a conceptual framework of I4.0 only 27 documents meet the selection criteria. This information was added in the methodology section. 

Reviewer 3 Report

The manuscript is well written and the topic is good and inline with the current research trend.

Author Response

REPLY TO REVIEWERS' COMMENTS

We are thankful to the reviewers for their important comments. By taking in account of their suggestions, some sections of the manuscript were modified for clarity (using the MS word track changes tool). Below, we give our rebuttals to the comment.

Reviewer 3

Comments and Suggestions for Authors

The manuscript is well written and the topic is good and in line with the current research trend.

Response to comment:

Thanks for your comment.

Reviewer 4 Report

Authors must make the following corrections in the paper:

- In section 1 (Introduction), authors must present the structure of the paper.
- Authors must explain better the academic contribution of the paper.

-Authors should develop the conclusions of the work and refer in more detail to the next steps of the work

Author Response

REPLY TO REVIEWERS' COMMENTS

We are thankful to the reviewers for their important comments. By taking in account of their suggestions, some sections of the manuscript were modified for clarity (using the MS word track changes tool). Below, we give our rebuttals to the comment.

Reviewer 4

Comments and Suggestions for Authors

Authors must make the following corrections in the paper:

Comment 1:

- In section 1 (Introduction), authors must present the structure of the paper.

Response to comment 1 :

Thanks for your comment. The manuscript was modified according to your suggestion.

Comment 2:

- Authors must explain better the academic contribution of the paper.

Response to comment 2:

The introduction part and also conclusions were modified to add the contributions of this work

Comment 3:

-Authors should develop the conclusions of the work and refer in more detail to the next steps of the work

Response to comment 3:

The conclusion section was modified by attending to your suggestions.

Reviewer 5 Report

  • This is interesting research that reviews the impacts of sustainability in Industry 4.0. It is clearly written and has important logistical issues.
  • It is recommended to better explain the methodology to be used in Fig. 2 as it mixes in each classification different topics. It would also be convenient to add the universe and sample to be studied and the criteria used to select the information from the papers.
  • In part 3, it is convenient to make a table and diagrams showing the information in an orderly fashion. There is a lot of information and it is not possible to identify what is the economic influence, for example what are the contributions made in each industry by country. This is requested in order to be able to make comparisons and have a greater analysis, for example, to answer which countries incorporate more sustainability? in which industries? in which dimensions? does this have to do with the level of development they have?. But to classify by some characteristic that helps to identify the dimensions.
  • In the paragraph on line 409, it mentions efficiency and effectiveness, but it is not made explicit how the economic aspects and customer satisfaction are related, or does it refer to effectiveness? associated with added value. 
  • In 3.3 it is also recommended to generate tables or figures to classify the information. This dimension also includes economic aspects, even though the classification is social. If separated by dimension, the social aspects would not have to consider the economic aspects.
  • If item 3 were further classified, more analysis could be generated in the discussion.
  • Table 1 does not show what the novelty of the business is.
  • Table 1 does not show what the novelty of the business is. It is advisable to add other characteristic(s) in the classification so that it is clear what the difference of the information in each paper is. To understand what is the importance for Industry 4.0.
  • Line 611 lacks justification for income targeting.
  • In the discussion and conclusions add more analysis.

Author Response

REPLY TO REVIEWERS' COMMENTS

We are thankful to the reviewers for their important comments. By taking in account of their suggestions, some sections of the manuscript were modified for clarity (using the MS word track changes tool). Below, we give our rebuttals to the comment.

Reviewer 5

Comments and Suggestions for Authors

Comment 1:

This is interesting research that reviews the impacts of sustainability in Industry 4.0. It is clearly written and has important logistical issues.

It is recommended to better explain the methodology to be used in Fig. 2 as it mixes in each classification different topics. It would also be convenient to add the universe and sample to be studied and the criteria used to select the information from the papers.

Response to comment 1:

Thanks for your valuable comments. We agree that the previous version of the manuscript has some issues in the methodology section. So, the section was modified by attending to your comment.

Comment 2:

In part 3, it is convenient to make a table and diagrams showing the information in an orderly fashion. There is a lot of information and it is not possible to identify what is the economic influence, for example what are the contributions made in each industry by country. This is requested in order to be able to make comparisons and have a greater analysis, for example, to answer which countries incorporate more sustainability? in which industries? in which dimensions? does this have to do with the level of development they have?. But to classify by some characteristic that helps to identify the dimensions.

Response to comment 2:

Thanks for your comments. Part 3 was modified by adding tables that include information about the case study, industry type, application, and country.

Comment 3:

In the paragraph on line 409, it mentions efficiency and effectiveness, but it is not made explicit how the economic aspects and customer satisfaction are related, or does it refer to effectiveness? associated with added value.

Response to comment 3:

The term effectiveness is related to the equipment and the labour efficiency is related to reducing waste time (this was added to the text).

Comment 4:

In 3.3 it is also recommended to generate tables or figures to classify the information. This dimension also includes economic aspects, even though the classification is social. If separated by dimension, the social aspects would not have to consider the economic aspects.

Response to comment 4:

Section 3.3 was modified according to your suggestions.

Comment 5:

If item 3 were further classified, more analysis could be generated in the discussion.

Response to comment 5:

Section 3.3 was modified according to your suggestions and discussion was added.

Comment 6:

Table 1 does not show what the novelty of the business is.

Table 1 does not show what the novelty of the business is. It is advisable to add other characteristic(s) in the classification so that it is clear what the difference of the information in each paper is. To understand what is the importance for Industry 4.0.

Response to comment 6:

The case study presented in table 1 (now table 4) was previously disccused in each dimension and information of industry type, application and country was added.

Comment 7:

Line 611 lacks justification for income targeting.

Response to comment 7:

Information was added.

Comment 8:

In the discussion and conclusions add more analysis.

Response to comment 8:

The conclusion part was modified by adding justifications, implications and future work.

Round 2

Reviewer 1 Report

Authors included all comments from my side, manuscript should be published.

Reviewer 5 Report

The information in the manuscript is well organized and the classifications and how sampling is performed are better understood.